# Adjuvant Therapy for Melanoma: Past, Current, and Future Developments

**DOI:** 10.3390/cancers12071994

**Published:** 2020-07-21

**Authors:** Alessandro A. E. Testori, Silvia Chiellino, Alexander C.J. van Akkooi

**Affiliations:** 1Department of Dermatology, Fondazione IRCCS Policlinico San Matteo, 27100 Pavia, Italy; 2Department of Medical Oncology, Fondazione IRCCS Policlinico San Matteo, 27100 Pavia, Italy; s.chiellino@smatteo.pv.it; 3Department of Surgical Oncology, Netherlands Cancer Institute–Antoni van Leeuwenhoek, 1066cx Amsterdam, The Netherlands; a.v.akkooi@nki.nl

**Keywords:** adjuvant therapy, melanoma, anti PD-1, target therapy, immunotherapy

## Abstract

This review describes the progress that the concept of adjuvant therapies has undergone in the last 50 years and focuses on the most recent development where an adjuvant approach has been scientifically evaluated in melanoma clinical trials. Over the past decade the development of immunotherapies and targeted therapies has drastically changed the treatment of stage IV melanoma patients. These successes led to trials studying the same therapies in the adjuvant setting, in high risk resected stage III and IV melanoma patients. Adjuvant immune checkpoint blockade with anti-CTLA-4 antibody ipilimumab was the first drug to show an improvement in recurrence-free and overall survival but this was accompanied by high severe toxicity rates. Therefore, these results were bypassed by adjuvant treatment with anti-PD-1 agents nivolumab and pembrolizumab and BRAF-directed target therapy, which showed even better recurrence-free survival rates with more favorable toxicity rates. The whole concept of adjuvant therapy may be integrated with the new neoadjuvant approaches that are under investigation through several clinical trials. However, there is still no data available on whether the effective adjuvant therapy that patients finally have at their disposal could be offered to them while waiting for recurrence, sparing at least 50% of them a potentially long-term toxic side effect but with the same rate of overall survival (OS). Adjuvant therapy for melanoma has radically changed over the past few years—anti-PD-1 or BRAF-directed therapy is the new standard of care.

## 1. Introduction

Historically melanoma was not broadly treated worldwide with adjuvant therapy. This was largely due to the absence of effective systemic therapy options. Interferon (IFN) was studied in many different schedules (high-dose, intermediate-dose, low-dose, pegylated-IFN, with or without induction, or shorter or longer maintenance therapy), but in general the effect was minimal [1,2,3,4,5,6,7,8,9,10,11]. Other approaches have also been attempted with vaccines or anti-angiogenic agents (VEGFi).

This situation changed in 2010 with the success of immune checkpoint blockade (ICB) with anti-CLTA-4 and anti-PD-1 antibodies, and the simultaneous results of selective inhibition of the MAP kinase pathway with BRAF and MEK inhibitors. These therapies (first as monotherapies, later also as combinations) first showed safety and clinical activity in unresectable stage III and metastatic stage IV patients [12,13,14,15,16,17,18,19,20], which led to the Food and Drug Administration (FDA) and European Medicines Agency (EMA) approvals of these drugs. The following step was to test these newly-approved agents as adjuvant therapy for high-risk resected stage III melanoma.

In this paper, we will discuss the history of adjuvant therapy for melanoma using interferon, vaccines, and anti-angiogenic agents, as well as the current developments with adjuvant immunotherapy (IT) and targeted therapy (TT) and finally, we will discuss some future approaches. During the last few decades, several revisions of the American Joint Committee on Cancer (AJCC) staging system have been proposed to reach the 8th edition published at the end of 2017. Until the finding of really efficacious treatments during the last 10 years, the real impact of the staging system represented more an academic then a clinical issue. Since then the impact of significant changes in the classification represent an important aspect to be considered; from the AJCC 7th to the 8th edition, the subdivision of four categories of risk of stage III may have caused some confusion in the selection of the correct patient population to be proposed for an adjuvant treatment, since most of the trials which have resulted in the success of the new therapies were AJCC-7th-edition-based. Clinicians should keep this in mind before addressing a prescription to all patients, specifically those with a limited risk of relapse.

## 2. What Is the Problem? How Big Is the Risk? What Is the Appropriate Patient Group to Target?

Before we continue to discuss the outcome of different adjuvant systemic therapy approaches, it is important to realize which group of patients is eligible for such treatments. There is no clear definition of what constitutes high-risk melanoma. Until now, classic staging with histopathological and clinical factors has been used to classify patient prognosis. For melanoma, the most-used staging worldwide has been that of the AJCC. According to the 7th edition of the AJCC staging the 5-year survival of stage III melanoma ranges from 78% for IIIA to 40% for IIIC [21]. Following the new subdivision of stage III proposed in the 8th edition, prognosis appears to present a wider range, where stage IIIA has a favorable prognosis at 5 years of 93%, stage IIIB has a prognosis of 83%, stage IIIC has a prognosis of 69%, and stage IIID has a prognosis of 32%. It is unclear exactly what percentage of new melanoma cases will develop into stage III disease in due time. Estimation is that approximately 15–25% of all these new cases will progress sooner or later into stage III disease. Obviously, the risk of this happening is lower for a pT1a melanoma (<5%) compared to a pT4b melanoma (40–50%). Luckily, the overwhelming amount of newly-diagnosed melanoma is pT1 (72%) [22,23].

Stage IIIB/C will be considered by most to be high risk, but stage IIIA with a 5-year survival rate of 78% might not be considered as high risk by all. Paradoxically, according to the 7th edition of the AJCC staging system, stage IIB/C had a worse prognosis than IIIA. This is likely due to the fact that patients needed not to be mandatorily staged by sentinel node (SN) procedure and thus they were ‘clinically’ staged as stage I/II, but actually had undiagnosed microscopic disease in the regional node(s), therefore N+ disease was likely missed.

Even so, the prognosis within stage IIIA melanoma is also heterogeneous. Many different parameters have been tested, but sentinel node (SN) tumor burden according to the Rotterdam criteria seems to be the most reproducible and robust way to determine prognosis within stage III, where OS ranges from 91% for <1 mm metastases to 57% for metastases of 1 mm and larger [24,25,26].

The Rotterdam criteria were first identified in a cohort of 262 stage I/II patients who underwent a SN, where 77 patients had metastases in their sentinel nodes. This was validated by a multicenter Eurpean Organization for Research and Treatment of Cancer (EORTC) study of 1088 SN+ patients from nine European sites. Finally, the 1 mm threshold for high-risk SN+ disease was confirmed by an independent cohort of 350 patients from Australia. At the same time, Murali et al. [27] did a study that tested the interobserver variability of the different ways to assess SN-tumor burden (Rotterdam criteria, Starz classification, microanatomic location, etc.) and demonstrated that the diameter of the largest lesion was the most reproducible, but the clinical relevance of melanoma micrometastases (<0.1 mm) in sentinel nodes is an important, interesting, and unclear topic—are these nodes to be considered negative or at least of no clinical relevance, or it is correct to consider these patients as stage III? Therefore, this threshold was used in the EORTC 18071, COMBI-AD, and EORTC 1325 adjuvant studies as inclusion criteria for stage IIIA disease [21,22,23].

The new, 8th edition of AJCC staging, has somewhat complicated things, as now stage III is divided into four subcategories rather than three. The 5-year survival of stage IIIA disease (regardless of SN-tumor burden) is now 93%, whereas IIB and IIC now have a 5-year survival of 87% and 82%, respectively. Thus, it seems logical that the new studies, discussed further below, will target stage IIB/C melanoma patients for adjuvant systemic therapy [28].

At the same time, most will recognize the fact that classic staging with histopathological and clinical factors will become obsolete once high-level molecular biomarkers are able to reliably predict prognosis. For instance, there are already a few panels of gene expression profiles (GEP) being tested for this purpose [29,30,31].

## 3. The Past: History of Adjuvant Therapy in Melanoma

Probably the first adjuvant scientific approach proposed on melanoma patients was the subcutaneous inoculation of bacille Calmette–Guérin vaccine (BCG) after completion of surgery, both after resection of high-risk primary and for nodal metastatic patients. The World Health Organization (WHO) conducted a retrospective analysis of AJCC stage—melanoma patients were included in a trial describing a significantly-prolonged disease-free survival (DFS) and overall survival (OS) *p* = 0.0006 and 0.007, respectively), in the subgroup of patients who received BCG but only if the patients converted from a negative to a positive reaction to BCG during treatment [32].

These findings where not confirmed by other studies with similar designs and in any case the comparative arm with chemotherapy (usually dacarbazine (DTIC)) never added any clinical benefit to the results [33,34].

The next generation trials were based on the combination of biochemotherapy and interferon (IFN) proposed at different schedules of administration in terms of duration and dosages, mostly high or intermediate doses. The background of biochemotherapy proposed in the adjuvant setting, derived from the experience in advanced disease, where even if no OS improvement had been ever obtained, an increased benefit was shown in terms of response rates. The biochemotherapy regimens included platinum-derived chemotherapeutics, IFN and interleukin-2 (IL-2). Despite a higher grade of toxicity, no clinical benefit was shown versus the single-arm IFN-based approach, thus such an approach has been abandoned since last century when studies were completed or stopped due to futility or low accrual [35].

## 4. Interferon

IFN alfa effect is mediated by T-helper-1 (T_h_1) antitumor responses, but also inhibits the proliferation of melanoma cells. It decreases intracellular and secretory levels of vascular endothelial growth factor (VEGF) in melanoma cell lines reducing the vascular neoangiogenesis around the tumor’s growth. Tumor immunogenicity is enhanced by IFN alfa administration promoting the rise of anti-tumor immunity as confirmed by several studies on the major histocompatibility complex (MHC) class I expression.

The milestone of IFN adjuvant treatment in melanoma is represented by the Eastern Cooperative Oncology Group (ECOG) 1684 trial, since has formed the basis for the approval (first by FDA in 1995, then in Europe) of the first and, for a long time, the only approved adjuvant drug in melanoma patients. IFN was studied worldwide before and after the results of ECOG 1684. Different regimens were proposed based on toxicity issues (it is better to have a low toxicity profile, but to keep patients under treatment for longer periods), where low dosages were administered, or, on the contrary, to impact with very high dosages like the endo-venous-infused induction phase of the ECOG trial (chemotherapy-like-based concept), repeated a few times at a schedule of every other month for four cycles.

A tentative approach to combining the proposed antiangiogenic effect of IFN, based on the persistency of drug availability in circulation, with the intermediate–high dose efficacy concept was the background of the European Organization for Research and Treatment of Cancer (EORTC) 18991 trial where pegylated IFN was the study drug. Pegylation permits a slow diffusion of the drug in the body after a weekly administration of the drug in the subcutis. The quality of life of patients was more acceptable compared to most of the alternatives proposed with a standard IFN formulation, but the results did not confirm a major benefit from the point of view of OS and DFS.

Several meta-analyses have been proposed on the numerous clinical trials that, over approximately 30 years, have been proposed on IFN. Globally, a benefit of circa 3% in OS is the best result that can be reasonably proposed to patients when offering them any schedule on IFN adjuvant settings. Some countries still have IFN as the only approved drug in the adjuvant setting, and this is the reason we dedicate such a long paragraph to this drug—the correct selection of patients to be treated with IFN may still be an important topic if the new drugs are not available. In these countries the choice to be discussed is related to the schedule to be proposed, since the most frequently-approved regimen is still the ECOG-1684-based dosage. Unfortunately this is the most toxic since more than 70% of patients cannot reach the completion of treatment during one year due to high levels of important toxicity (grade 3 and 4). The suggestion is to try to propose a form of dosage which succeeds to keep patients under treatment for a reasonable period of months, reducing the initial dose in case such a dose reduction could not be proposed from the beginning of the treatment due to a rigid formal necessity to use the approved original dosage of IFN [1,2,3,4,5,6,7,8,9,10,11,12].

Within the EORTC IFN trials, a selection of patients for whom IFN may be more indicated has been conducted initially through a retrospective analysis of past studies, and finally on a dedicated study on ulcerated primary melanoma patients. The EORTC 18081 study, despite the fact that the accrual was never completed for the predefined study power, confirmed the retrospective evaluations and the results on EORTC 18991 study on patients with ulcerated melanoma (where N+ < 1 mm nodal metastases also reached a benefit). These results have been very recently published; at a 3.4-year median follow-up, the estimated HR, stratified by Breslow thickness and localization of the primary tumor, for the PEG-IFNα-2b group compared to the observation group was 0.66 (95% CI: 0.32–1.37), and the 3-year relapse free survival (RFS) rate was 80.0% (95% CI: 65.7–88.8%) and 72.9% (95% CI: 58.3–83.0%), respectively. Distant metastases free survival (DMFS) was prolonged—HR, 0.39 (95% CI: 0.15–0.97), and the 3-year DMFS rate was 90.6% (95% CI: 78.9–96.0%) vs. 76.4% (95% CI: 62.1–85.9%). The EORTC 18081 PEG-IFNα-2b randomized trial, observed a similar HR (0.69) for RFS to the previous EORTC trials (0.69) indicating that in ulcerated primary melanoma patients (even including patients with small micrometastases in the sentinel node), IFN can be the treatment of choice to be proposed in countries where there is no reimbursement yet for the new adjuvant immunotherapies or target therapies [36].

Finally, the era of IFN could have a new development since some studies described the correlation of prognosis in patients developing autoantibodies. If the development of autoimmune antibodies in melanoma patients treated with IFN could be confirmed, this could be used as a prognostic marker as well as in the selection of patients to be treated for a longer duration. Autoantibodies develop after a variable period after the start of IFN—some after three months, some after one year. The EORTC collected samples from patients from the EORTC 18991 study (peg-intron) but the presence of autoantibodies could not be correlated with any clinical benefit of the patients. Anticardiolipin, antithyroglobulin, and antinuclear antibodies were analyzed before and during the 5-year follow-up as well as during and after the conclusion of IFN administration. Only patients with a baseline negative expression of autoimmunity were selected. Of these, 18% of the 220 patients admitted in the collateral study evaluation arm versus 52% in the IFN arm developed autoantibodies, indicating that autoimmune antibodies are neither a prognostic nor a predictive marker for selecting patients with better outcome [37].

## 5. Vaccines

The identification of potential antitumor responses driven by neoantigens derived from tumor specific mutation has led to a large variety of research projects; since most mutated proteins are essentially tumor-specific, a vaccination strategy can potentially be used in clinical practice to create a specific antitumoral response.

The EORTC proposed the EORTC 18961 trial, where patients were randomized to receive the GM2 ganglioside (a tumor-specific antigen present in the majority of melanoma cells) vs. observation in 1314 high-risk primary melanoma patients (thickness > 1.5 mm, without metastases). After the second interim analysis at 1.8 years of follow up, the trial was stopped due to futility regarding RFS (HR 1.00; *p* = 0.99) and the fact that it was detrimental in terms of OS (HR 1.66; *p* = 0.02). These findings evidenced the potential role of tolerance activation against tumoral cells, with a clear modification of the concept being that an anti-tumoral vaccine is not a simple approach to be offered to patients where the concept is to administer a treatment that is of low toxicity. If it had worked it could have been a success, but in a case of inefficacy, at least it did not worsen the situation. This trial demonstrated that a vaccine, even if not toxic, can be detrimental on the prognosis of melanoma patients [38].

The Glaxo-Smith-Kline (GSK) DERMA trial was a phase 3, double-blind, randomized (2:1) trial that compared, in patients with completely resected, stage IIIB or IIIC, MAGE-A3-positive cutaneous melanoma, a treatment with 13 intramuscular injections of recombinant MAGE-A3 combined with an immunostimulant versus a placebo, over a 27-month period. Human melanoma antigen A3 (MAGE A3) is a tumor-specific protein identified in approximately 60% of melanoma cells. This study failed to show efficacy of the MAGE-A3 injections in terms of disease-free survival [39]. Following this study, a selection of patients based on a gene signature identified to potentially predict efficacy of the vaccine was studied. The negative results of such a sophisticated analysis have most probably brought to end any further development of MAGE A3 research on melanoma patients, at least as a monotherapy, without a combination with any new drugs (such as immune checkpoint blockade) [40].

## 6. Anti-Angiogenic

The most relevant adjuvant trial based on antiangiogenesis is the AVAST-M study conducted in the UK. This randomized phase 3 trial compared, in 1343 patients with resected stage IIB, IIC, and III cutaneous melanoma, adjuvant bevacizumab (7.5 mg/kg i.v. 3-weekly for one year) to standard observation. Bevacizumab (Avastin, F. Hoffman-La Roche AG, Basel, CH) is a recombinant humanized monoclonal antibody to vascular endothelial growth factor (VEGF) approved for different cancers and studied with some preliminary results on advanced melanoma. The primary end point was detection of an 8% difference in 5-year overall survival rate; secondary end points included disease-free interval (DFI). Angiogenesis is a key point in the development of cancer progression and VEGF has been found over-expressed in melanoma and correlated with worsened prognosis. Even if adjuvant bevacizumab showed an advantage in terms of DFI (51% vs. 45% on observation (HR 0.85; 95% CI 0.74–0.99, *p* = 0.003)), OS at five years was 64% for both groups, the study failed to show an advantage in terms of survival and this strategy cannot be recommended in clinical practice [41].

## 7. The Current Landscape, New Drugs: Immunotherapy (IT) and Target Therapy (TT)

The first adjuvant trial of the new era was the EORTC 18071 trial, which randomized patients between an induction phase of four courses of high dose ipilimumab (IPI) (10 mg/kg), followed by maintenance every 12 weeks for three years versus placebo. This is probably the first, last, and only clean adjuvant study, which was not influenced (much or at all) by post-relapse treatments, since those were not yet (widely) available for most patients in this study. Thus, it demonstrated a consistent RFS, distant-metastasis-free survival (DMFS) and OS effect in favor of IPI, albeit accompanied with a high rate of severe toxicity [42,43].

The ECOG 1609 study (NCT01274338) was designed initially as a two-arm study comparing adjuvant IFN with high dose ipililumab (IPI) (10 mg/kg), but was later amended to include a third arm, when the now-more-common stage IV dose of ipilimumab (3 mg/kg) was introduced. The study showed that ipilimumab 3 mg/kg had a significant OS benefit (HR 0.78). There was no difference in terms of RFS between the two ipilimumab arms. However, the frequency of grade 3/4 treatment-related adverse events (AEs) was significantly less for the 3 mg/kg dose and therefore the need to discontinue treatment due to toxicity was less for this lower dose [44,45].

These adjuvant IPI developments have since been bypassed by more recent result. First, the Checkmate 238 randomized resected stages IIIB/C, and stage IV (AJCC 7th) between high dose IPI or nivolumab (3 mg/kg) every two weeks for one year. This study showed a significant benefit of 10% in absolute RFS benefit at 12 months, with a minimum follow-up of 18 months. This effect was maintained in a recent update and one needs to be reminded that the hazard ratio of 0.65 (0.51–0.83, *p* < 0.001) was versus a proven active comparator rather than to placebo. The rate of grade 3/4 AEs was also much lower at 14.4% vs. 45.9%, making this a more attractive adjuvant therapy, in terms of both AEs and efficacy [46].

As a matter of fact, IPI has never been processed for approval in Europe, being adopted only by FDA-related countries.

Another study with an adjuvant anti-PD-1 agent was the EORTC 1325/Keynote 054 study, which randomized stage IIIA (>1 mm), IIIB, and IIIC to either a year of 3-weekly fixed dose pembrolizumab 200 mg or placebo [47]. A recent update of this study by Eggermont et al. at ASCO 2020 showed the 3-year RFS rates were 63.7% vs. 44.1% [48].

The amount of grade 3/4 treatment-related AEs was 14.7%. Interestingly, this study is the only phase 3 adjuvant RCT to include a cross-over design. Patients in the placebo arm could switch to the pembrolizumab treatment after relapse and patients who were ≥6 months since last pembrolizumab could be re-treated. This study will thus answer the question as to whether it is better to give pembrolizumab early, as an adjuvant approach, which means to all stage III patients, or if one can still salvage patients with later treatment after disease progression, saving at least 50% of them, already cured by surgery, from an unnecessary treatment.

Finally, there have been two studies reported on adjuvant BRAF-directed therapy. The first was a study with a BRAF inhibitor alone (vemurafenib). Interestingly, this study also included stage IIC patients. However, monotherapy with BRAF inhibitors is no longer routine for stage IV disease since the combination of BRAF + MEK inhibition is more effective and less toxic. Perhaps therefore, this BRIM 8 study failed to meet the pre-planned statistical plan, as it did show RFS benefit for stage IIC/IIIA/IIIB patients, but none for the stage IIIC subcategory, which had to be positive to be able to consider the total study positive [49].

Moreover, at the same time, the Roche COMBI-AD study, which randomized BRAF V600E/K-positive IIIA (>1 mm), IIIB, and IIIC disease patients between combination dabrafenib and trametinib versus a double placebo, did show a significant benefit for RFS. The first interim analysis showed a trend towards OS benefit and an updated follow-up showed a maintained (albeit lesser) effect even four years after commencing study participation [50]. A recent update at ASCO 2020 demonstrated a maintained RFS benefit after five years of 52% vs. 36%, but also a DMFS benefit of 11% at 5-year (65% vs. 54%) [51] (Table 1).

## 8. Neo Adjuvant Studies

Last, but not least, there have also been some neoadjuvant studies with both targeted and immunotherapy agents. The philosophy behind the neoadjuvant concept in oncology has become stronger in the last few years. From gastro-intestinal locally-advanced cancers the benefit of surgery, when proposed after a medical and/or radiation approach, appears clear. In melanoma this represents a completely new topic since the operability concept has rarely been the main reason for proposing a medical versus a surgical approach. The background on melanoma is mainly based on the success of the therapies in advanced disease, and it appears logical to propose the same medical treatment that can render a stage IV patient disease-free in an earlier situation in presence of locoregional disease only. Surgery will be kept as a salvation approach in case of failure of the new therapy. To reach this goal most studies are designed to study the pathologic locoregional disease on all patients who first undergo the medical approach by a surgical excision, so that the concept of offering the standard approach can be preserved for them. Targeted therapies have been shown to be able to rapidly reduce tumor load and easy surgical complete resections, although they are not likely to create a durable effect if the systemic therapy is not continued post-surgery [52,53,54,55]. The Bristol-Myers Squibb (BMS) OpACIN trial was a randomized phase-IB/II trial were Patients were randomly assigned to receive ipilimumab at 3 mg/kg plus nivolumab at 1 mg/kg, either in four adjuvant courses, or to receive the same doses split into two neoadjuvant plus two adjuvant courses postoperatively. The trial was not powered or designed to look at response rates, but nevertheless, a very high overall response rate (ORR) of 80% was unexpectedly seen after two neoadjuvant courses, with a large proportion (60%) of patients achieving a (near)-complete pathologic response (pCR). The trial was designed to look at feasibility and safety, which demonstrated that it was feasible, but highly toxic, with 90% of patients developing grade 3/4 adverse events and most discontinuing treatment after only two to three courses. Another endpoint of the study was the expansion of tumor-specific T-cells, which interestingly showed a larger expansion of both known as newly-detected tumor-specific T-cell clones in the neo-adjuvant cohort compared to the adjuvant cohort [56,57]. More importantly, it demonstrated that only two courses of ‘low dose’ ipilimumab (1 mg/kg) + nivolumab (3 mg/kg) had far fewer grade 3/4 adverse events (20%). This study was expanded with the personalized response-driven adjuvant therapy after combination ipilimumab and nivolumab in high-risk resectable stage III melanoma (PRADO) trial expansion cohort, whose results are still not published.

In the PRADO, patients will no longer all undergo a complete therapeutic lymph node dissection, but first undergo an excision of the index node. Based on the pathologic response found in the index node, the patients with a (near)-pCR (maximum 10% viable tumor) will undergo follow-up, those with a partial response (10–50% viable tumor) will undergo a node dissection, but continue with follow-up thereafter and the non-responders (>50% viable tumor) will undergo a node dissection, followed by adjuvant radiotherapy and adjuvant systemic therapy with either BRAF/MEK inhibition (only for BRAF-mutated melanomas) or anti-PD-1.

Recently, at ASCO 2020, the first result of the PRADO demonstrated an ORR of 71% with a (near)-pCR rate of 61%, this meant that 59/99 patients did not need to undergo a complete therapeutic lymph node dissection. Follow-up will determine if the omission of the removal of the remaining lymph nodes was detrimental to patients [58].

Another study by Amaria et al. which randomized neo-adjuvant nivolumab (3 mg/kg) to neo-adjuvant combined ipilimumab (3 mg/kg) + nivolumab (1 mg/kg) showed similar results with an ORR of 73% for the combination (45% achieving pCR), but again at the cost of frequent grade 3/4 toxicity (73%). Both the ORR, pCR, and toxicity was much higher than the neo-adjuvant nivolumab monotherapy cohort [59].

A study by Huang et al. [60] examined a single dose of pembrolizumab prior to surgery and found that 29.6% already achieved a pCR, which is similar to the rate found by Amaria et al. One might question if the response rate could not be higher if the neo-adjuvant treatment was prolonged. However, the Amaria et al. study did not seem to indicate this. Moreover, the combination with ipilimumab seems more promising, considering the higher response rates (include pCR rates). Finally, it is planned to start an international randomized phase III trial (so far called “Nadina study”), which will compare two courses of neo-adjuvant ‘low dose’ ipilimumab (1 mg/kg) + nivolumab (3 mg/kg) followed by surgery and then a year of anti-PD-1 adjuvant systemic therapy (Table 2).

## 9. The Future

Ongoing studies include the Checkmate 915 study (NCT03068455), which randomizes patients between nivolumab (3 mg/kg) and combination low-dose ipilimumab (1 mg/kg) and nivolumab (3 mg/kg) and is not expected to report first results until the end of 2020.

The SWOG 1404 study randomizes patients between IFN and pembrolizumab, with a third arm being added later when FDA approves high dose ipilimumab. It will report first results by 2020.

Other ongoing adjuvant trials include two trials targeting stage IIB/C melanoma. The first is the Keynote-716 trial (NCT03553836), which randomizes stage IIB/C melanoma patients between 17 cycles of 200 pembrolizumab every three weeks versus placebo. The second is the CheckMate76K (NCT04099251), which randomizes stage IIB/C melanoma patients between adjuvant nivolumab and placebo. Both trials are expected to first report results by 2023–2024.

Finally, the EORTC 1902 study is the first and only study examining adjuvant sequential treatment with BRAF/MEK inhibitors encorafenib and binimetinib followed by anti-PD-1 versus anti-PD-1 alone versus placebo. This study will start accrual in 2020. 

Table 3 lists most of the future studies that are currently at the discussion phase of discussion. A precise date of activation of these trials will need to be confirmed.

## 10. Conclusions

Adjuvant systemic therapy for melanoma has radically changed over the recent past [61]. Adjuvant IFN should be reserved only for patients with ulcerated melanomas and no metastases or small metastases in the SN in those countries where there is no reimbursement for the newer adjuvant therapies.

The main endpoint used to define the efficacy of adjuvant therapy has been relapse-free survival (RFS). Although overall survival (OS) is the most clinically-meaningful end-point, it has the disadvantage of requiring a long follow-up and being influenced by subsequent treatment lines, especially in the adjuvant setting. An end-point that is achieved faster like RFS could potentially expedite approval of a new drug, furthermore, in the specific setting an endpoint that is not influenced by subsequent treatments provides clearer and cleaner information.

The issue is whether a benefit in terms of an endpoint with these characteristics expresses an advantage in terms of survival.

To answer this question, is in 2018 a meta-analysis was conducted by Suciu et al. The aim was to assess if RFS is a valid surrogate for OS among resected stage II–III melanoma patients. This study showed that RFS appeared to be a valid surrogate end-point for OS for adjuvant randomized studies assessing interferon or a checkpoint inhibitor [62,63].

Most stage III trials have been proposed based on the prognostic risk of the AJCC 7th edition. The 2017 8th edition had divided stage III in four risk categories, presenting a larger range of disease progression (from 7% on IIIA to 78% on IIID). N1a patients presenting a SNB-detected node with a Rotterdam criteria of minimal tumor bulk of less than 1 mm in maximum diameter and a thin primary melanoma will present a better prognosis compared to a thick primary melanoma patient with no signs of stage III disease. Most studies have excluded these patients from adjuvant trials and should be kept off treatment.

For BRAF wild type, the current new standard of care is anti-PD-1 (nivolumab and pembrolizumab). For BRAF-mutated melanomas, both combined BRAF and MEK inhibition and anti-PD-1 are considered viable options, as there is no comparative data (yet) to select either drug in stage III-IV completely-resected melanoma patients, so at the moment the treatment is selected by the clinician on the basis of each individual patient’s characteristic.

Therefore, the treatment toxicity profile (also related to the patient’s professional activity and/or his comorbidities) together with the route of administration (oral continuous vs. intravenous) are the most reasonable elements to determine the treatment choice. With respect to toxicity, we have to also consider that discontinuation rate is higher with targeted therapy than with immunotherapy (25% vs. 6–8%), but targeted therapy toxicity can be easily managed by stopping administration while immune-related adverse events may persist after discontinuation and may require long-lasting systemic treatments [50].

The last frontier will be the setting of neoadjuvant approaches. It is too early to indicate if this approach will be confirmed as standard-of-care in the presence of operable metastatic disease, but international studies are ongoing on this topic.

A final consideration is related to an open question arising during a multidisciplinary discussion on the therapeutic options at relapse after adjuvant therapy. The possibility of treating patients with high-risk melanoma in an adjuvant setting makes it necessary to think in perspective and to choose a treatment also considering a hypothetical therapeutic sequence.

Regardless of the treatment chosen, adjuvant therapy reduces the risk of recurrence in a percentage ranging between 30% and 50%. Therefore, treatment in an advanced setting will not be necessary for a reasonable number of patients. In addition, the longer the time to relapse observed in patients receiving previous adjuvant treatment, the more reasonable it is to propose, at recurrence, a therapeutic re-challenge with the same drug. Clearly, the issue of re-challenge is particularly relevant for BRAF WT patients for whom immunotherapy is the only option.

For BRAF MT patients, that can benefit from both strategies, the need to understand the best therapeutic sequence in the transition from adjuvant to metastatic setting is mandatory and needs a prospective evaluation. The concept from which we can start to design a specific trial is probably the concept of immunomodulation by targeted therapy. Growing evidence suggests that BRAF inhibitors, in addition to their direct anti-tumor activity, can also promote immune responses to melanoma. This is because BRAF signaling has an immunosuppressive effect and blocking his pathway can lead to increase in antigen expression, T-cell activity, and PDL1 expression. So, in a hypothetical therapeutic sequence, it must be considered that molecular target therapy can make cancer cells more exposed and therefore more sensitive to an immunotherapeutic approach [64,65,66].

## Figures and Tables

**Table 1 cancers-12-01994-t001:** Summary of the current landscape of adjuvant trials.

Trial	Design	AJCC 7th Edition Stages	HR RFS	HR DMFS	HR OS
Immunotherapy (IT)					
EORTC 18071	IPI 10 mg/kg vs. placebo	IIIA (SN > 1 mm), IIIB, IIIC No in-transits	0.76	0.76	0.72
EORTC 1325	Pembrolizumab 200 mg vs. placebo	IIIA (SN > 1 mm), IIIB, IIIC No in-transits	0.57	0.53	NA
Checkmate 238	IPI 10 mg/kg vs. Nivo 3 mg/kg	IIIB, IIIC, IV In-transits allowed	0.65	0.73	NA
ECOG 1609	IPI 10 mg/kg vs. IPI 3 mg/kg vs. HD IFN-α2b	IIIB, IIIC, IV M1a-b	1.0	NA	NA
Targeted Therapy (TT)					
BRIM-8	Vemurafenib vs. Placebo	IIC, IIIA (SN > 1 mm), IIIB, IIIC In-transits allowed	0.54 (IIC-IIIB) 0.8 (IIIC)	NA	NA
COMBI-AD	Dabrafenib + trametinib vs. Placebo	IIIA (SN > 1 mm), IIIB, IIIC In-transits allowed	0.47	0.51	0.57

HR: hazard ratio.

**Table 2 cancers-12-01994-t002:** Summary of neo-adjuvant trial developments.

Trial	Regimen	N	pCR %	RFS (mo)	FU (mo)
Targeted Therapy (TT)					
Amaria et al.	Dabrafenib + trametinib	21	58	19.7	18.6
Long et al.	Dabrafenib + trametinib	35	49	23.0	27.0
Immunotherapy (IT)					
Blank et al.	Ipi + Nivo	10	33	NR	32
Amaria et al.	Nivo Ipi + Nivo	12 11	25 45	NR NR	20
Huang et al.	Pembrolizumab	30	19	NR	18
Rozeman et al.	Ipi + Nivo	86	57	NR	8.3

FU: follow up.

**Table 3 cancers-12-01994-t003:** Ongoing/planned adjuvant trials.

Trial	Regimen	Population (AJCC 8)	N	Expected
Adjuvant				
Targeted Therapy (TT)				
EORTC 1902	Encorafenib + binimetinib vs. Placebo	IIA, IIB, IIC	800	Accrual starts 2020
Adjuvant				
Immunotherapy (IT)				
Checkmate 915	Ipi 1 mg/kg + Nivo 3 mg/kg vs. Nivo 3 mg/kg	IIIB, IIIC, IIID, IV	1943	10/2020
SWOG 1404	Pembrolizumab 200 mg vs. Ipi 10 mg/kg	IIIA (N2a), IIIB, IIIC, IV	1378	10/2020
Keynote 716	Pembrolizumab 200 mg vs. placebo	IIB, IIC	954	10/2022
CA209-76K	Nivolumab 480 mg vs. placebo	IIB, IIC	1000	01/2024

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
