# Peer review of "Adjuvant Therapy for Melanoma: Past, Current, and Future Developments"

_cancers, 2020, doi:10.3390/cancers12071994_

Round 1
Reviewer 1 Report
As the authors described, the adjuvant therapy for melanoma has radically changed over the past years. Therefore, it is very meaningful to review the past progress and focus on the most recent development. This review is well written, however there are some points to be answered.
(major points)
- The authors described that many different parameters have been tested, but SN tumor burden according to the Rotterdam criteria seems to be the most reproducible and robust way to determine prognosis within stage III. If so, the authors should explain the data to prove it.
- The authors described about the AJCC staging of 7th and 8th Please mention about the problem that most clinical trials have been conducted by the AJCC 7th. So, evidences are consisted from AJCC 7th. However, the clinicians have started to use AJCC 8th at the bedside. Caution should be stated about this problem.
- The authors should mention about the personalized mRNA vaccines trials.What kind of trials are going on? How was known about the effect?
(minor points)
- It seems that the manuscript needs careful editing. For example;(page)…but in general the effect was d minimal. (page 5) …toxicity was less for the mg/kg dose.
- Abbreviations should be defined when first used in the manuscript. For example; (page 2) SN
Author Response
- dear reviewer all your questions have been answerd in the text. here you can find the sections modified in the text
- question
- The authors described that many different parameters have been tested, but SN tumor burden according to the Rotterdam criteria seems to be the most reproducible and robust way to determine prognosis within stage III. If so, the authors should explain the data to prove it.
- answer
The Rotterdam criteria were first identified in a cohort of 262 stage I/II patients that underwent a SN, 77 patients had metastases in their sentinel nodes. This was validated by a multicenter EORTC study in 1088 SN+ patients from 9 European sites. Finally, the 1 mm threshold for high risk SN+ disease was confirmed by an independent cohort of 350 patients from Australia. At the same time, Murali et al. did a study that tested the interobserver variability of the different ways to assess SN tumor burden (Rotterdam criteria, Starz classification, microanatomic location, etc.) and demonstrated that the diameter of the largest lesion was the most reproducible, but the clinical relevance of melanoma micrometastases (<0.1 mm) in sentinel nodes is an important, interesting and unclear topic: are these nodes to be considered negative or at least of no clinical relevance or it is correct to consider these patients as stage III?
question
The authors described about the AJCC staging of 7th and 8th Please mention about the problem that most clinical trials have been conducted by the AJCC 7th. So, evidences are consisted from AJCC 7th. However, the clinicians have started to use AJCC 8th at the bedside. Caution should be stated about this problem. - answer
During the last decades several revisions of the American Joint Committee on Cancer (AJCC) staging system have been proposed to reach the 8th edition published at the end of 2017. Until the finding of really efficacious treatments during the last 10 years, the real impact of the staging system represented more an academic then a clinical issue. Since then the impact of significant changes on the classification can represent an important aspect to be considered: from the AJCC 7th to the 8th edition, the subdivision on 4 categories of risk of stage III may bring some confusion in the selection of the correct patients population to be proposed for an adjuvant treatment, since most of the trials which have resulted in the success of the new therapies were AJCC 7th edition based. Clinicians should keep this concept on mind before addressing a prescription to all patients, specifically to those with a limited risk of relapse. The authors should mention about the personalized mRNA vaccines trials.What kind of trials are going on? How was known about the effect?
(minor points)
- It seems that the manuscript needs careful editing. For example;(page)…but in general the effect was d minimal. (page 5) …toxicity was less for the mg/kg dose. ok done
- Abbreviations should be defined when first used in the manuscript. For example; (page 2) SN ok done
Reviewer 2 Report
Testori and Colleagues have thoroughly reviewed the adjuvant landscape of melanoma treatment. Below are some minor suggestions they may wish to consider
Suggestions
- The section on Risk quotes the AJCC 7th Given the 8th addition has been available for over 1 yr I would suggest quoting statistics from this and then comment/acknowledge the landmark studies were recruited under the 7th addition, not the other way around.
- -particularly so with the discussion of IIB/C this is simplified in AJCC 8 where stage II is on the basis of a -ve SNBx
- The discussion of IFN is long and detailed, a reader may think the authors are suggesting this may be an option for some pts? If this is not the case I would suggest shortening and using it to give historical perspective only
- Modern era studies
- -? discuss analysis based on AJCC 8th
- -would be great to update based on recent ASCO data (no need for further review post this)
- I suggest separating the neoadj discussion into a separate section
-starting with the perceived benefits/risk of Neoadj and then a discussion of the data
-great to add PRADO data from ASCO
- Other topics the authors may consider discussing:
- -RFS as an endpoint (particularly for IO trials where prolonged disease control/’cure’ is possible in those with advanced disease
- Pros/cons of PD1 vs DAB/Tram in BRAF mt pts
- Disease behaviour/treatment options at relapse
Author Response
- dear reviewer all your questions have been answerd in the text. here you can find the sections modified in the text
- question
- The section on Risk quotes the AJCC 7th Given the 8th addition has been available for over 1 yr I would suggest quoting statistics from this and then comment/acknowledge the landmark studies were recruited under the 7th addition, not the other way around.-particularly so with the discussion of IIB/C this is simplified in AJCC 8 where stage II is on the basis of a -ve SNBx
- answer
-
During the last decades several revisions of the American Joint Committee on Cancer (AJCC) staging system have been proposed to reach the 8th edition published at the end of 2017. Until the finding of really efficacious treatments during the last 10 years, the real impact of the staging system represented more an academic then a clinical issue. Since then the impact of significant changes on the classification can represent an important aspect to be considered: from the AJCC 7th to the 8th edition, the subdivision on 4 categories of risk of stage III may bring some confusion in the selection of the correct patients population to be proposed for an adjuvant treatment, since most of the trials which have resulted in the success of the new therapies were AJCC 7th edition based. Clinicians should keep this concept on mind before addressing a prescription to all patients, specifically to those with a limited risk of relapse. Following the new subdivision of stage III proposed on the 8th edition, prognosis appears to present a wider range, where stage IIIA has a favourable prognosis at 5 years of 93%, stage IIIB 83%, stage IIIC 69% and stage IIID 32%. Even so, the prognosis within stage IIIA melanoma is heterogeneous too. Many different parameters have been tested, but sentinel node (SN) tumor burden according to the Rotterdam criteria seems to be the most reproducible and robust way to determine prognosis within stage III, where OS ranges from 91% for < 1 mm metastases to 57% for metastases of 1 mm and larger.
-
- question
- The discussion of IFN is long and detailed, a reader may think the authors are suggesting this may be an option for some pts? If this is not the case I would suggest shortening and using it to give historical perspective only
- answer
- Some countries still have IFN as the only approved drug in the adjuvant setting, and this is the reason we dedicate such a long paragraph to this drug: the correct selection of patients to be treated with IFN may still be an important topic if the new drugs are not available.Modern era studies
- question
- -? discuss analysis based on AJCC 8th
- -answer
- Most stage III trials have been proposed based on the prognostic risk of AJCC 7th edition. The 2017 8th edition have divided stage III in 4 risk categories, presenting a larger range of disease progression (from 7% on IIIA to 78% on IIID). N1a patients presenting a SNB detected node with a Rotterdam criteria of minimal tumor bulky less then 1 mm in maximum diameter and a thin primary melanoma will present a better prognosis compared to a thick primary melanoma patient with no signs of stage III disease. Most studies have excluded these patients from adjuvant trials and should be kept off treatment.
- would be great to update based on recent ASCO data (no need for further review post this) ok done
- question
- I suggest separating the neoadj discussion into a separate section
-starting with the perceived benefits/risk of Neoadj and then a discussion of the data
answer
Neo adjuvant studies
Last, but not least, there have also been some neoadjuvant studies with both targeted and immunotherapy agents. All the philosophy behind the neoadjuvant concept in oncology is becoming stronger and stronger in the last few years. From gastro-intestinal locally advanced cancers it appears clearly the benefit that surgery may reach when proposed after a medical and/or radiation approach. In melanoma this represents a completely new topic, since the operability concept has been rarely the main reason to propose a medical versus a surgical approach. The background on melanoma is mainly based of the success of the therapies in advanced disease, and it appears logical to propose the same medical treatment that can render a stage IV patient disease free in an earlier situation in presence of locoregional disease only. Surgery will be kept as a salvation approach in case of failure of the the new therapy. To reach this goal most studies are designed to study the pathologic locoregional disease on all patients undergoing first the medical approach by a surgical excision, so that the concept of offering the standard approach can be preserved to them.
question
-great to add PRADO data from ASCO
answer
The OpACIN trial was a randomized phase IB/II trial that compared 2 courses of neo-adjuvant combination ipilimumab (3 mg/kg) + nivolumab (1 mg/kg) followed by surgery and thereafter 2 more courses after surgery to upfront surgery and 4 courses postoperatively. The trial was not powered or designed to look at response rates, but nevertheless, a very high overall response rate (ORR) of 80% was unexpectedly seen after 2 neo-adjuvant courses, with a large proportion (60%) of patients achieving a (near)-complete pathologic response (pCR). The trial was designed to look at feasibility and safety, which demonstrated that it was feasible, but highly toxic with 90% of patients developing grade 3/4 adverse events and most discontinuing treatment after only 2-3 courses. Another endpoint of the study was the expansion of tumor specific T-cells, which interestingly showed a larger expansion of both known as newly detected tumor specific T-cell clones in the neo-adjuvant cohort compared to the adjuvant cohort52-53. More importantly, it demonstrated that only 2 courses of ‘low dose’ ipilimumab (1 mg/kg) + nivolumab (3 mg/kg) had far less grade 3/4 adverse events (20%). This study was expanded with the PRADO expansion cohort, whose results are still not published. In the PRADO, patients will no longer all undergo a complete therapeutic lymph node dissection, but first undergo an excision of the index node. Based on the pathologic response found in the index node, the patients with a (near)-pCR (max 10% viable tumor) will undergo follow-up, those with a partial response (10-50% viable tumor) will undergo a node dissection, but continue with follow-up thereafter and the non-responders (>50% viable tumor) will undergo a node dissection, followed by adjuvant radiotherapy and adjuvant systemic therapy with either BRAF/MEK inhibition (only for BRAF mutated melanomas) or anti-PD-1. Interestingly, this study is the only phase 3 adjuvant RCT to include a cross-over design. Patients in the placebo arm could switch to the pembrolizumab treatment after relapse and patients who were ≥ 6 months since last pembrolizumab could be re-treated. This study will thus answer the question if it is better to give pembrolizumab early, as an adjuvant approach, which means to all stage III patients, or if one can still salvage patients with later treatment after disease progression, saving at least 50% of them, already cured by surgery, from an unnecessary treatment. Finally, there have been 2 studies reported on adjuvant BRAF directed therapy. The first was a study with a BRAF inhibitor alone (vemurafenib). Interestingly, this study also includes stage IIC patients. However, monotherapy with BRAF inhibitors is no longer routine for stage IV disease, since the combination of BRAF + MEK inhibition is more effective and less toxic. Perhaps therefore, this BRIM 8 study failed to meet the pre-planned statistical plan, as it did show benefit of stage IIC/IIIA/IIIB patients for RFS, but none for the stage IIIC subcategory, which had to be positive to be able to consider the total study positive29. Moreover, at the same time, the COMBI-AD study, which randomized BRAF V600E/K positive IIIA (> 1 mm), IIIB, IIIC disease patients between combination dabrafenib and trametinib versus a double placebo, did show a significant benefit for RFS. The first interim analysis showed a trend towards OS benefit and an updated follow-up showed a maintained (albeit lesser) effect even 4 years after commencing study participation48 . A recent update at ASCO 2020 demonstrated a maintained RFS benefit after 5-years of 52 vs. 36%, but also a DMFS benefit of 11% at 5-year (65% vs. 54%). Tab 1).
- Other topics the authors may consider discussing:
- question
- -RFS as an endpoint (particularly for IO trials where prolonged disease control/’cure’ is possible in those with advanced disease
- answer
-
The main endpoint used to define the efficacy of adjuvant therapy has been relapse free survival (RFS). Although overall survival (OS) is the most clinically meaningful end-point, it has the disadvantage of requiring a long follow-up and being influenced by subsequent treatment lines, especially in the adjuvant setting. An end-point that is achieved faster like RFS could potentially expedite approval of a new drug, furthermore, in the specific setting an endpoint that is not influenced by subsequent treatments, provides clearer and cleaner information. The issue is whether a benefit in terms of an endpoint with these characteristics expresses an advantage in terms of survival. To answer this question, in 2018 a meta-analysis was conducted by Suciu et al. The aim was to assess if RFS is a valid surrogate for OS among resected stage II–III melanoma patients. This study showed that RFS appeared to be a valid surrogate end-point for OS for adjuvant randomized studies assessing interferon or a checkpoint inhibitor.
- question
- Pros/cons of PD1 vs DAB/Tram in BRAF mt pts
- answer
For BRAF wild type, the current new standard of care is anti-PD-1 (nivolumab, pembrolizumab). For BRAF mutated melanomas, both combined BRAF&MEK inhibition or anti-PD-1 are both considered viable options, as there is no comparative data (yet) to select either drug in stage III-IV completely resected melanoma patients, so at the moment the treatment is selected by the clinician on the basis of each individual patient’s characteristic.
Clearly, BRAF mutated patients can benefit from both adjuvant strategies (i.e., combination of dabrafenib plus trametinib and immunotherapy with nivolumab or pembrolizumab), there is no scientific evidence that can guide the choice because of the lack of prospective head-to-head comparison trials, so at the moment the treatment is selected by the clinician on the basis of each individual patient’s characteristic.
Therefore, treatment toxicity profile (also related to the patient's professional activity and/or his comorbidities) together with the route of administration (oral continuous vs intravenous) are the most reasonable elements to determine the treatment choice. About toxicity we have also to consider that discontinuation rate is higher with targeted therapy than with immunotherapy (25% vs 6-8%), but targeted therapy toxicity can be easily managed by stopping administration while immune-related adverse events may persist after discontinuation and may require long lasting systemic treatments.
- question
- Disease behaviour/treatment options at relapse
- answer
-
A final consideration is related to an open question arising during a multidisciplinary discussion on the therapeutic options at relapse after adjuvant therapy. The possibility of treating patients with high-risk melanoma in an adjuvant setting makes it necessary to think in perspective and to choose a treatment also considering a hypothetical therapeutic sequence.
Regardless the treatment chosen, adjuvant therapy reduces the risk of recurrence in a percentage ranging between 30 and 50%. Therefore, treatment in advanced setting will not be necessary for a reasonable number of patients. In addition, the longer the time to relapse observed in patients receiving previous adjuvant treatment can re-propose a therapeutic re-challenge with the same drug reasonable at recurrence. Clearly, the issue of re-challenge is particularly relevant for BRAF WT patients in whom immunotherapy is the only option.
For BRAF MT patients, that can benefit from both strategies, the need to understand the best therapeutic sequence in the transition from adjuvant to metastatic setting is mandatory and needs a prospective evaluation. The concept from which we can start to design specific trial is probably the concept of immunomodulation by targeted therapy. Growing evidence suggests that BRAF inhibitors, in addition to their direct anti-tumor activity, can also promote immune responses to melanoma. This is because BRAF signaling has an immunosuppressive effect and blocking his pathway can lead to increase in antigen expression, T-cell activity and PDL1 expression. So, in a hypothetical therapeutic sequence, it must be considered that molecular target therapy can make cancer cells more exposed and therefore more sensitive to an immunotherapeutic approach.
Round 2
Reviewer 1 Report
The authors did not respond one of my question.
'The authors should mention about the personalized mRNA vaccines trials.
What kind of trials are going on? How was known about the effect?'
Author Response
Comments and Suggestions for Authors
Testori and Colleagues have thoroughly reviewed the adjuvant landscape of melanoma treatment. Below are some minor suggestions they may wish to consider
Suggestions
- question
- The section on Risk quotes the AJCC 7th Given the 8th addition has been available for over 1 yr I would suggest quoting statistics from this and then comment/acknowledge the landmark studies were recruited under the 7th addition, not the other way around.
- -particularly so with the discussion of IIB/C this is simplified in AJCC 8 where stage II is on the basis of a -ve SNBx
- answer
- During the last decades several revisions of the American Joint Committee on Cancer (AJCC) staging system have been proposed to reach the 8th edition published at the end of 2017. Until the finding of really efficacious treatments during the last 10 years, the real impact of the staging system represented more an academic then a clinical issue. Since then the impact of significant changes on the classification can represent an important aspect to be considered: from the AJCC 7th to the 8th edition, the subdivision on 4 categories of risk of stage III may bring some confusion in the selection of the correct patients population to be proposed for an adjuvant treatment, since most of the trials which have resulted in the success of the new therapies were AJCC 7th edition based. Clinicians should keep this concept on mind before addressing a prescription to all patients, specifically to those with a limited risk of relapse.
Stage IIIB/C will be considered by most to be high risk, but stage IIIA with a 5-year survival rate of 78% might not be considered as high risk by all. Paradoxically, according to the 7th edition of AJCC staging system, stage IIB/C had a worse prognosis than IIIA. This is likely due to the fact that patients needed not to be mandatorily staged by SN procedure and thus they were ‘clinically’ staged as stage I/II, but actually had undiagnosed microscopic disease in the regional node(s), therefore N+ disease was likely missed.
Even so, the prognosis within stage IIIA melanoma is heterogeneous too. Many different parameters have been tested, but sentinel node (SN) tumor burden according to the Rotterdam criteria seems to be the most reproducible and robust way to determine prognosis within stage III, where OS ranges from 91% for < 1 mm metastases to 57% for metastases of 1 mm and larger 24-26.
- question
- The discussion of IFN is long and detailed, a reader may think the authors are suggesting this may be an option for some pts? If this is not the case I would suggest shortening and using it to give historical perspective only
- answer
-
The EORTC 18081 study, despite the fact that the accrual was never completed for the predefined study power, confirmed the retrospective evaluations and the results on 18991 patients with ulcerated melanoma (where also N+ < 1 mm nodal metastases reached a benefit). These results have been very recently published: at a 3.4-year median follow-up, the estimated HR, stratified by Breslow thickness and localisation of the primary tumor, for the PEG-IFNα-2b group compared to the observation group was 0.66 (95% CI: 0.32-1.37), and the 3-year RFS rate was 80.0% (95% CI: 65.7-88.8%) and 72.9% (95% CI: 58.3-83.0%), respectively. DMFS was prolonged: HR, 0.39 (95% CI: 0.15-0.97), and the 3-year DMFS rate was 90.6% (95% CI: 78.9-96.0%) vs 76.4% (95% CI: 62.1-85.9%). The EORTC 18081 PEG-IFNα-2b randomised trial, observed a similar HR (0.69) for RFS as the previous EORTC trials (0.69) indicating that in ulcerated primary melanoma patients (even including patients with small micrometastases in the sentinel node), IFN can be the treatment of choice to be proposed in countries where there is no reimbursement yet for the new adjuvant immunotherapies or target therapies36.
- Modern era studies
- question
- -? discuss analysis based on AJCC 8th
- answer
- see above answer to question 1
- question
- -would be great to update based on recent ASCO data (no need for further review post this)
- answer
- all the adj protocols updated at ASCO 20 have been adjourned in the text
- question
- I suggest separating the neoadj discussion into a separate section
- answer
- done
question
-starting with the perceived benefits/risk of Neoadj and then a discussion of the data
answer
Last, but not least, there have also been some neoadjuvant studies with both targeted and immunotherapy agents. All the philosophy behind the neoadjuvant concept in oncology is becoming stronger and stronger in the last few years. From gastro-intestinal locally advanced cancers it appears clearly the benefit that surgery may reach when proposed after a medical and/or radiation approach. In melanoma this represents a completely new topic, since the operability concept has been rarely the main reason to propose a medical versus a surgical approach. The background on melanoma is mainly based of the success of the therapies in advanced disease, and it appears logical to propose the same medical treatment that can render a stage IV patient disease free in an earlier situation in presence of locoregional disease only. Surgery will be kept as a salvation approach in case of failure of the the new therapy. To reach this goal most studies are designed to study the pathologic locoregional disease on all patients undergoing first the medical approach by a surgical excision, so that the concept of offering the standard approach can be preserved to them. Targeted therapies have shown to be able to rapidly reduce tumor load and easy surgical complete resections, although they are not likely to create a durable effect if the systemic therapy is not continued post-surgery52-55
question
-great to add PRADO data from ASCO
answer
done
- Other topics the authors may consider discussing:
- question
- -RFS as an endpoint (particularly for IO trials where prolonged disease control/’cure’ is possible in those with advanced disease
- answer
-
The main endpoint used to define the efficacy of adjuvant therapy has been relapse free survival (RFS). Although overall survival (OS) is the most clinically meaningful end-point, it has the disadvantage of requiring a long follow-up and being influenced by subsequent treatment lines, especially in the adjuvant setting. An end-point that is achieved faster like RFS could potentially expedite approval of a new drug, furthermore, in the specific setting an endpoint that is not influenced by subsequent treatments, provides clearer and cleaner information.
The issue is whether a benefit in terms of an endpoint with these characteristics expresses an advantage in terms of survival.
To answer this question, in 2018 a meta-analysis was conducted by Suciu et al. The aim was to assess if RFS is a valid surrogate for OS among resected stage II–III melanoma patients. This study showed that RFS appeared to be a valid surrogate end-point for OS for adjuvant randomized studies assessing interferon or a checkpoint inhibitor63-64.
- question
- Pros/cons of PD1 vs DAB/Tram in BRAF mt pts
- answer
-
For BRAF mutated melanomas, both combined BRAF&MEK inhibition or anti-PD-1 are both considered viable options, as there is no comparative data (yet) to select either drug in stage III-IV completely resected melanoma patients, so at the moment the treatment is selected by the clinician on the basis of each individual patient’s characteristic.
Clearly, BRAF mutated patients can benefit from both adjuvant strategies (i.e., combination of dabrafenib plus trametinib and immunotherapy with nivolumab or pembrolizumab), there is no scientific evidence that can guide the choice because of the lack of prospective head-to-head comparison trials, so at the moment the treatment is selected by the clinician on the basis of each individual patient’s characteristic.
Therefore, treatment toxicity profile (also related to the patient's professional activity and/or his comorbidities) together with the route of administration (oral continuous vs intravenous) are the most reasonable elements to determine the treatment choice. About toxicity we have also to consider that discontinuation rate is higher with targeted therapy than with immunotherapy (25% vs 6-8%), but targeted therapy toxicity can be easily managed by stopping administration while immune-related adverse events may persist after discontinuation and may require long lasting systemic treatments65.
- question
- Disease behaviour/treatment options at relapse
- answer
-
A final consideration is related to an open question arising during a multidisciplinary discussion on the therapeutic options at relapse after adjuvant therapy. The possibility of treating patients with high-risk melanoma in an adjuvant setting makes it necessary to think in perspective and to choose a treatment also considering a hypothetical therapeutic sequence.
Regardless the treatment chosen, adjuvant therapy reduces the risk of recurrence in a percentage ranging between 30 and 50%. Therefore, treatment in advanced setting will not be necessary for a reasonable number of patients. In addition, the longer the time to relapse observed in patients receiving previous adjuvant treatment can re-propose a therapeutic re-challenge with the same drug reasonable at recurrence. Clearly, the issue of re-challenge is particularly relevant for BRAF WT patients in whom immunotherapy is the only option.
For BRAF MT patients, that can benefit from both strategies, the need to understand the best therapeutic sequence in the transition from adjuvant to metastatic setting is mandatory and needs a prospective evaluation. The concept from which we can start to design specific trial is probably the concept of immunomodulation by targeted therapy. Growing evidence suggests that BRAF inhibitors, in addition to their direct anti-tumor activity, can also promote immune responses to melanoma. This is because BRAF signaling has an immunosuppressive effect and blocking his pathway can lead to increase in antigen expression, T-cell activity and PDL1 expression. So, in a hypothetical therapeutic sequence, it must be considered that molecular target therapy can make cancer cells more exposed and therefore more sensitive to an immunotherapeutic approach66-68.
